# Hepatoblastoma—The Evolution of Biology, Surgery, and Transplantation

**DOI:** 10.3390/children6010001

**Published:** 2018-12-21

**Authors:** Irene Isabel P. Lim, Alexander J. Bondoc, James I. Geller, Gregory M. Tiao

**Affiliations:** 1Division of General and Thoracic Surgery, Cincinnati Children’s Hospital Medical Center, 3333 Burnet Ave, Cincinnati, OH 45229, USA; ireneisabel.lim-beutel@cchmc.org (I.I.P.L.); alex.bondoc@cchmc.org (A.J.B.); 2Division of Oncology, Cancer and Blood Diseases Institute, Cincinnati Children’s Hospital Medical Center, 3333 Burnet Ave, Cincinnati, OH 45229, USA; james.geller@cchmc.org

**Keywords:** hepatoblastoma, liver, cancer, transplant

## Abstract

The most common primary malignant liver tumor of childhood, hepatoblastoma has increased in incidence over the last 30 years, but little is still known about its pathogenesis. Discoveries in molecular biology provide clues but have yet to define targeted therapies. Disease-free survival varies according to stage, but is greater than 90% in favorable risk populations, in part due to improvements in chemotherapeutic regimens, surgical resection, and earlier referral to liver transplant centers. This article aims to highlight the principles of disease that guide current treatment algorithms. Surgical treatment, especially orthotopic liver transplantation, will also be emphasized in the context of the current Children’s Oncology Group international study of pediatric liver cancer (AHEP-1531).

## 1. Introduction

Hepatoblastoma (HB) is the most common primary liver tumor in the pediatric population. The incidence of this tumor is 1.5 cases/million population per year (approximately 1% of cancers for young children) but has been increasing over the last 30 years. Its incidence has increased by as much as 2.7% per year [1], and this phenomenon may be related to improved survival of premature infants, as very low birth weight has been associated with HB [2]. In the United States, approximately 250 children are diagnosed with HB each year. There is an association with Beckwith–Wiedemann syndrome, familial adenomatous polyposis, and trisomy 18, though the majority of cases are spontaneous [3]. Few studies have identified a clear mechanistic pathway of this tumor. Current treatment is guided by the degree of local tumor burden as determined by the pre-treatment and post-treatment extent of disease (PRETEXT or POST-TEXT) system [4]. While conventional surgical resection remains the cornerstone of management, advances in chemotherapy regimens and orthotopic liver transplant offer additional therapies in otherwise unresectable cases. Using an evidence-based risk stratification treatment algorithm, current 5-year event-free survival (EFS) rates for patients with lower risk HB approach 80% [5] but may be significantly worse (30–40%) for relapsed or higher risk HB. This review will focus on the current biologic principles of disease and the treatment algorithms available, with a particular emphasis on the current success and challenges of orthotopic liver transplant (OLT) in HB.

## 2. Biology

As an embryonal tumor, HB presents with morphologic features encompassing diverse cell types with varying degrees of differentiation. Tumors with fetal histology and low mitotic activity are associated with a favorable clinical course, while higher risk tumors tend to display significant atypia and anaplasia. The majority of HBs are epithelial types: pure fetal, embryonal, macrotrabecular, and small-cell undifferentiated (SCUD).

Unlike adult hepatocellular carcinoma (HCC), HB develops in the absence of liver disease. Previous studies have revealed a paucity of genetic alterations in these tumors overall. Rather, these tumors have a relatively stable genome [6] with only few pathways implicated. Up to 90% of tumors have aberrations in the Wnt signaling pathway [7]. A recent study using whole-exome sequencing showed that HB tumors contain few coding mutations, with younger patients more likely to have tumors with fewer somatic mutations [8]. More than 80% of tumors tested positive for mutations and deletions of *CTNNB1* [8], a key downstream component of the Wnt signaling pathway. *CTNNB1* alterations were the most frequent mutation observed in another study [9]. Most of these are interstitial deletions or missense point mutations in exon 3, resulting in β–catenin protein that cannot be degraded [7]. Clinical applications of the implied role of the Wnt pathway in HB, however, have yet to be determined.

While alpha-fetoprotein (AFP) level has been used as a prognostic indicator, no serum biomarker completely correlates with disease severity or relapse. AFP is elevated in up to 80–90% of patients with HB, but it can also be elevated in patients with malignant teratomas and yolk sac tumors. Significantly high or low serum AFP was associated with adverse prognosis [8]. Serum protein markers were analyzed using mass spectrometry, and of the target proteins isolated, Apo A-I was found to have increased expression in the HB subgroups compared to normal [10]. In addition, DNA methylation patterns may be used as predictors of outcome. One study compared methylation levels of the *AFP* gene in tumor versus normal liver tissue and found that tumor had lower *AFP* methylation levels [11]. Another study also found increased methylation of tumor suppressors in patients with metastatic tumors [12], suggestive of a role for predicting prognosis.

## 3. Clinical Presentation

Children typically present with an asymptomatic abdominal mass, with no associated systemic symptoms. High-quality cross-section imaging is paramount in the diagnosis, risk stratification, and thus treatment, of HB. Triple-phase computed tomography or magnetic resonance imaging with eovist, a gadolinium agent that is preferentially taken up by hepatocytes [13], are recommended. By ensuring adequate imaging, staging can be done correctly. All three major, international multicenter trial groups, the Children’s Oncology Group (COG), the International Society of Pediatric Oncology—Epithelial Liver Tumor Group (SIOPEL), and the Japanese Study Group for Pediatric Liver Tumors (JPLT), use the previously mentioned PRETEXT scheme to stratify tumors. Each patient is assigned a PRETEXT group depending on the number of contiguous uninvolved sections of the liver. Unlike Couinaud segments, sections divide the liver by the hepatic and portal veins [4]. There are four sections of the liver (Figure 1): left lateral (Couinaud segments 2 and 3), left medial (segments 4a and 4b), right anterior (segments 5 and 8), and right posterior (segments 6 and 7).

Tumors are further subclassified based on extension into key extra-parenchymal structures: M for distant metastatic disease, P for portal bifurcation or main portal vein involvement, V for involvement of all three hepatic veins or the inferior vena cava, F for multifocal tumor burden, C for caudate involvement, and E for extrahepatic extension into a contiguous intraabdominal organ. Seemingly straightforward, the PRETEXT system can be difficult to apply for some cases, particularly those with the “P” and “V” subclassifications (Table 1). The COG central review aims to further stratify these subclassifications, but for the purposes of the current AHEP-1531 study, P and V status is either negative or positive. P-positive tumors either obliterate or encase the main portal vein or both first-order portal veins, or tumor thrombus is present in either or both right and left portal veins or the main portal vein [4]. Similarly, V-positive tumors either obliterate or encase all three first-order hepatic veins or the intrahepatic inferior vena cava, or tumor thrombus is present in any or more first-order hepatic vein or the intrahepatic inferior vena cava [4]. Notably, PRETEXT has been shown to predict prognosis [14] but not resectability. The same classification system is used to evaluate tumors after chemotherapy for the POSTTEXT score after each cycle of chemotherapy.

## 4. Pathology

Determination of histological subtype is becoming increasingly important as it can change the treatment algorithm. Patients with a liver mass require adequate biopsy to not only get information on subtype diagnosis, but also to allow for chemotherapy response. Biopsies can be performed as percutaneous core, laparoscopic core or wedge, or open biopsies. The choice of biopsy method should carefully weigh the risk of bleeding with obtaining an adequate amount of tissue in the safest and least invasive approach. Open biopsy will ensure enough viable tissue for analysis but is associated with the greatest risk of bleeding. On the other hand, percutaneous biopsy is the least invasive method with a decreased, but still present, bleeding risk but multiple cores should be taken to ensure sufficient tissue. A minimum of three cores should be obtained, but when possible, five cores of tumor and one core of normal liver are recommended to allow for accurate histologic workup. Immunohistochemistry stains are also utilized, though their clinical importance varies. Alpha fetoprotein, Hep-Par1, and integrase interactor 1 (INI1) are just a few of the stains used. INI1-negative SCUD epithelial HB with low AFP may indicate the need for an alternative chemotherapy regimen, as these tumors are rhabdoid in origin [15]. CD44, an important surface marker molecule in various cancer stem cells, has been shown to have significantly higher expression in Stage III and IV tumors compared to Stage I and II [16]. Moreover, aberrant expression of CD44 as well as CD90 and CD133 contributes to disease progression, and thus decreased survival, in HB [17]. Ongoing research is directed towards the clinical implications of these histologic markers.

## 5. Subtypes

### 5.1. Well-Differentiated Fetal (WDF) Hepatoblastoma (HB)

Associated with better prognosis, well-differentiated fetal (WDF) HBs have uniformly small cells with a small central nucleus without a nucleolus. These cells demonstrate staining for glutamine synthetase, glypican 3 and membranous β-catenin [18]. Mitoses are rare, with rates of less than 2 per 10 high-powered fields [19]. This subtype is stratified as very low risk and treated with surgery only. While pure fetal HB with low mitotic activity occurs in less than 7% of patients [3], it is also associated with the best prognosis.

### 5.2. Embryonal HB

As its name suggests, embryonal HB corresponds to the embryonic stage of hepatocyte development. Mitoses are frequent and unlike WDF HB, the nucleus is large, with a prominent nucleolus.

### 5.3. Small Cell Undifferentiated (SCU) HB

Occurring in approximately 5% of HB cases, small cell undifferentiated (SCU) HB carries a poorer prognosis [19]. SCU HB has a small blue cell morphology with a high nuclear–cytoplasmic ratio and high mitotic rate [20]. Unlike WDF HB, SCU HB cells are negative for glutamine synthetase and glypican-3. Nuclear B-catenin is also frequently seen in SCU HB, which has been associated with lower EFS [21]. INI1 staining is an important part of risk stratification for this subtype, as loss of INI1 suggests a malignant rhabdoid origin [15] and thus, alternative management.

## 6. Treatment Algorithms

Until recently, treatment algorithms varied significantly as North American and European pediatric oncology consortium differed as to the role of surgical resection at diagnosis. COG guidelines recommended complete resection at diagnosis if possible, while SIOPEL studies typically utilized four cycles of chemotherapy prior to resection. All four study groups utilize cisplatin as the core chemotherapy agent. The standard COG regimen, also known as C5V, consists of cisplatin, 5-fluorouracil, and vincristine.

The COG AHEP-0731 trial utilized a stratified chemotherapy regimen, depending on the risk stratification. Very low-risk disease, such as PRETEXT I/II resected at diagnosis with pure fetal histology, does not receive chemotherapy. Pure fetal histology is not treated with chemotherapy as resection is curative [22]. PRETEXT I/II resected at diagnosis with any other histology is considered low-risk disease, for which C5V is recommended after surgery. Cases of intermediate risk, such as PRETEXT III/IV and SCUD histology, and high-risk disease (metastases and serum AFP at diagnosis less than 100 ng/ml) are given C5V with doxorubicin (C5VD). In addition, the COG AHEP-0731 study included evaluation of new agents in the treatment of high-risk HB. These patients received vincristine and irinotecan, and if found to respond, continued with additional two cycles of vincristine and irinotecan and six cycles of C5VD [23]. Those who did not have a good response received only six cycles of C5VD. HB responsiveness to chemotherapy in terms of residual tumor volume did not change significantly after the second chemotherapy cycle [24], and so tumor response was evaluated at this time. Vincristine and irinotecan were well tolerated and there was substantial activity against high-risk HB using this combination. Temsirolimus has since been added to vincristine and irinotecan for high-risk disease, and results of this addition are forthcoming.

The current AHEP-1531 study employs a novel risk stratification algorithm defined by the efforts of the CHIC collaborative (Figure 2). This risk stratification is based on the presence of positive annotation factors (V,P,E,F, or R), age (less than or greater than 8 years, or 3 years for PRETEXT IV), and AFP levels. Those in Group A, or the very low-risk group, receive cisplatin as long as the histology is not of the well-differentiated fetal type. Those in Group B, or the low-risk group, undergo two or four cycles of cisplatin, either as neoadjuvant or adjuvant therapy depending on resectability at presentation. Those of the intermediate-risk group, or Group C, are randomized to receive either cisplatin or C5VD before and after surgery. Those of Group D, or the high-risk group, undergo Société Internationale d’Oncologie Pédiatrique (SIOPEL) 4 induction, consisting of cisplatin and doxorubicin. Tumor response is assessed after blocks 2 and 3, and depending on clearance of metastases, patients are given carboplatin and doxorubicin or carboplatin and doxorubicin alternating with carboplatin and etoposide or vincristine and irinotecan.

## 7. Surgical Resection

Despite improvements in chemotherapy agents and protocols, treatment with intent for cure relies heavily on surgery. The timing and extent of surgical resection depend on the PRETEXT/POSTTEXT stage, response to neoadjuvant chemotherapy, and tumor biology. In the AHEP-0731 trial, guidelines on timing and extent of surgical resection are PRETEXT-based (Table 1). Upfront resection is only recommended for PRETEXT I and II tumors with no macrovascular involvement and for which resection can yield an at least 1-cm margin [13]. In the current AHEP-1531 trial, resection at diagnosis is only recommended for PRETEXT I and II tumors that are categorized as “very low” risk and for which there is an at least 1-cm radiographic margin of unaffected liver parenchyma between the tumor edge and the middle hepatic vein and portal bifurcation.

For other tumors, timing of resection is less straightforward (Figure 3). In the AHEP-1531 study, PRETEXT I and II tumors not resected at diagnosis should be biopsied and started on neoadjuvant chemotherapy. Resection can be considered after two cycles of chemotherapy for PRETEXT I/II low and intermediate risk, while high-risk tumors should be resected after induction block 3. PRETEXT III tumors, on the other hand, should all be biopsied at diagnosis and begun on neoadjuvant chemotherapy. PRETEXT III low- or intermediate-risk tumors are then re-imaged after chemotherapy cycle 2 and cycle 4. Patients with tumors deemed unresectable on imaging after the second chemotherapy cycle should be referred for OLT evaluation. Resection for PRETEXT III low risk can take place as early as after two cycles, with plans for two versus four cycles of adjuvant chemotherapy after surgery. PRETEXT III intermediate-risk tumors, however, should be considered for resection after the second cycle and no later than after the fourth cycle, with trisectionectomy or extreme resection often required (Figure 3). In one study of 20 patients, the majority of PRETEXT III and IV tumors achieved radiographic resectability after completion of two chemotherapy cycles [25], thus allowing for earlier resection and reduction in chemotherapy.

Patients presenting with PRETEXT IV tumors should receive early referral to a liver transplant center. Timeliness of referral does not necessarily mean a transplant is imminent, but rather, allows for completion of transplant workup. PRETEXT IV tumors are further classified as intermediate or high risk, with intermediate tumors resected after the fourth cycle. Patients with PRETEXT IV high risk require cross-sectional imaging after induction of block 3 of cisplatin chemotherapy. Those in whom metastases have cleared, or those older than 8 years of age without metastases, would be referred for surgical resection if local control is achievable or for transplant if the tumors are unlikely to be resectable after block 3. Transarterial chemoembolization has been reported as an adjunct to resection in these patients [26].

The most common location for distant HB metastases is the lung, with as many as 20% of HB patients presenting with pulmonary disease [27]. Computed tomography is the preferred method of identification and surveillance of pulmonary metastases. Patients in whom metastases persist should be considered for both surgical resection and pulmonary metastectomy if local control is feasible. If local control requires OLT, clearance of metastases with pulmonary metastectomy and induction chemotherapy are needed. Complete resection of pulmonary metastases, however, can be difficult, particularly if intraoperative identification of the nodule is suboptimal. The addition of indocyanine green (ICG) fluorescence has helped address this issue. Lesions as small as 0.062mm can be detected using ICG [28] (Figure 4). In addition to lung lesions, primary and recurrent tumor in the liver have also been accurately detected using ICG [29]. Pulmonary metastectomy after OLT for unresectable HB has been reported, also with the aid of ICG technology [30]. This clinical application presents a potential adjunct to surgery in the future. Of course, confirmation of lung clearance with cross sectional imaging is paramount prior to transplant.

Adequate margins for resection, while set to at least 1 cm in the AHEP-1531 study, are not clearly defined. Complete gross resection often involves a 3–5 mm rim of tissue around the tumor by electrocautery but a positive histo-pathologic margin may not be representative of the disease left behind [31]. Intraoperative ICG fluorescent imaging can be used to confirm clearance of tumor after both primary liver resection and pulmonary metastastectomy [28,29]. COG guidelines recommend a segmentectomy or lobectomy for PRETEXT I and II tumors when a minimum 1-cm margin is possible. In general, for PRETEXT I and II lesions, surgery can entail segmentectomy, sectionectomy, or even a hemihepatectomy to achieve adequate margins. Cases of PRETEXT III, or POST-TEXT I, II, or III with no V or P, often require lobectomy or trisectionectomy.

## 8. Liver Transplant

Pediatric OLT has evolved over the past two decades, particularly for HB. While the number of HB cases increased by 4-fold, OLT for HB has increased by more than 20-fold [32]. Similar to initial results of OLT for HCC with cirrhosis, liver transplant for HB in the 1990s resulted in high rates of recurrence and a mere 50% 2-year survival [33]. Improvements in chemotherapy and transplant expertise led to more favorable outcomes with the use of primary transplantation for unresectable HB [34]. One large single-center study of children who underwent OLT for HB reported a 10-year survival rate of 77.6% [35]. Rescue transplants had worse survival rates, with higher rates of tumor recurrence as compared to primary transplant for unresectable HB after chemotherapy [36]. In a Surveillance Epidemiology and End Results (SEER) database study from 1998 to 2009, overall five-year survival was equivalent for patients who underwent resection (85.6%) or OLT (86.5%) [37]. A more recent study found that OLT for unresectable malignant primary pediatric hepatic tumors, the majority of which were HB, had comparable outcomes to those undergoing transplant for nonmalignant causes [38]. At one transplant center, the ten-year overall survival was 84%, including patients who underwent salvage transplants and pulmonary metastectomy [39].

Early referral to liver transplant programs is crucial. Recurrence after OLT has been associated with PRETEXT IV lesions, longer waiting list time, and older age at time of transplant [40]. As disease recurrence is the most significant indicator of mortality [39], an early referral system that significantly reduces the time between tissue diagnosis and OLT [41] will optimize survival. In 2010, additional “exception” points were assigned to patients with HB in the pediatric end-stage liver disease (PELD) system. With this new criteria, patients with HB experience shorter waitlist time [42]. OLT is recommended in tumors with major venous invasion, multifocal PRETEXT IV and those with high likelihood of needing salvage transplant, whether due to positive margins or liver insufficiency. Piecemeal, nonanatomic resection of multifocal disease is not recommended. Patients with metastatic disease that cleared with chemotherapy or with pulmonary metastectomy are also eligible for transplant.

Complications of OLT for HB treatment are similar to those of OLT for other indications. These complications include hemorrhage, bile leak, and hepatic artery thrombosis. Rates, however, are higher in pediatric OLT than in adults, likely due to smaller vessels and use of segmental grafts [40]. For example, hepatic artery thrombosis was reported in as many as a quarter of pediatric liver transplant cases in one study [43]. Acute cellular rejection was also reported in the HB cohort [38]. Chemotherapy after OLT has also been shown to increase survival: overall 5-year survival was 86% for the chemotherapy group versus 62% for the non-chemotherapy group [38]. AFP levels with serial imaging are used to screen for recurrences, which remain the most common cause of death in this population.

## 9. Conclusions

HB treatment has evolved from a singular treatment plan to a risk-stratified management strategy. This change incorporates advances in the molecular biology of disease, novel chemotherapeutic regimens, and improvements in surgical techniques. In particular, improvements in transplant care and the recognition that OLT can provide primary rather than salvage treatment in HB are encouraging. Aggressive resection of pulmonary metastases has also contributed to the multimodal treatment of HB.

## Figures and Tables

**Figure 1 children-06-00001-f001:**
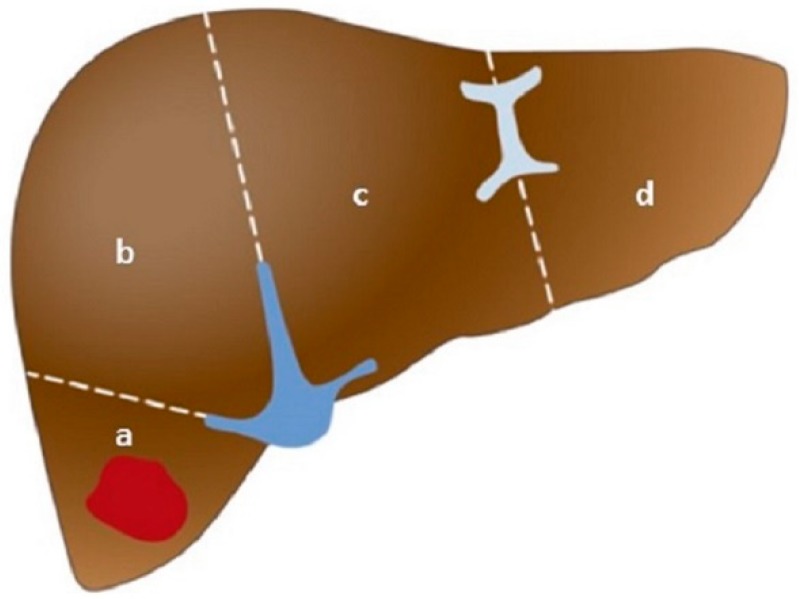
Liver sections [4]. The liver has four sections, separated by the hepatic and portal veins: (**a**) right posterior, (**b**) right anterior, (**c**) left medial, and (**d**) left lateral. Note that the caudate lobe is not considered part of these sections.

**Figure 2 children-06-00001-f002:**
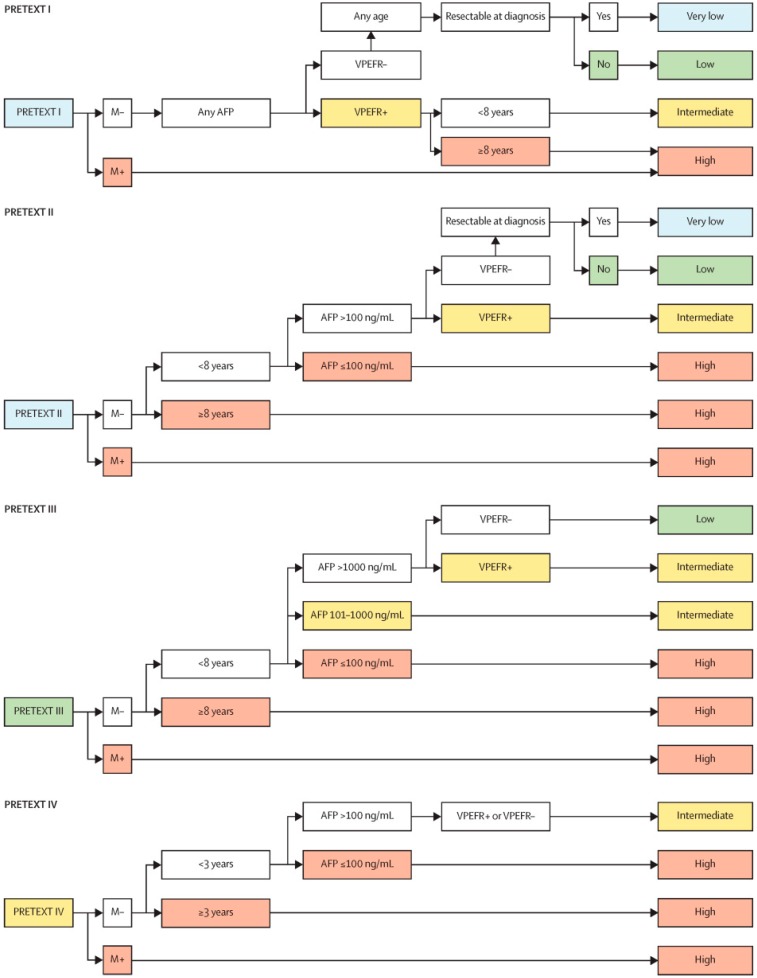
These stratification trees are used as the framework for management of hepatoblastoma in the AHEP-1531 study [5]. AFP = alpha-fetoprotein.

**Figure 3 children-06-00001-f003:**
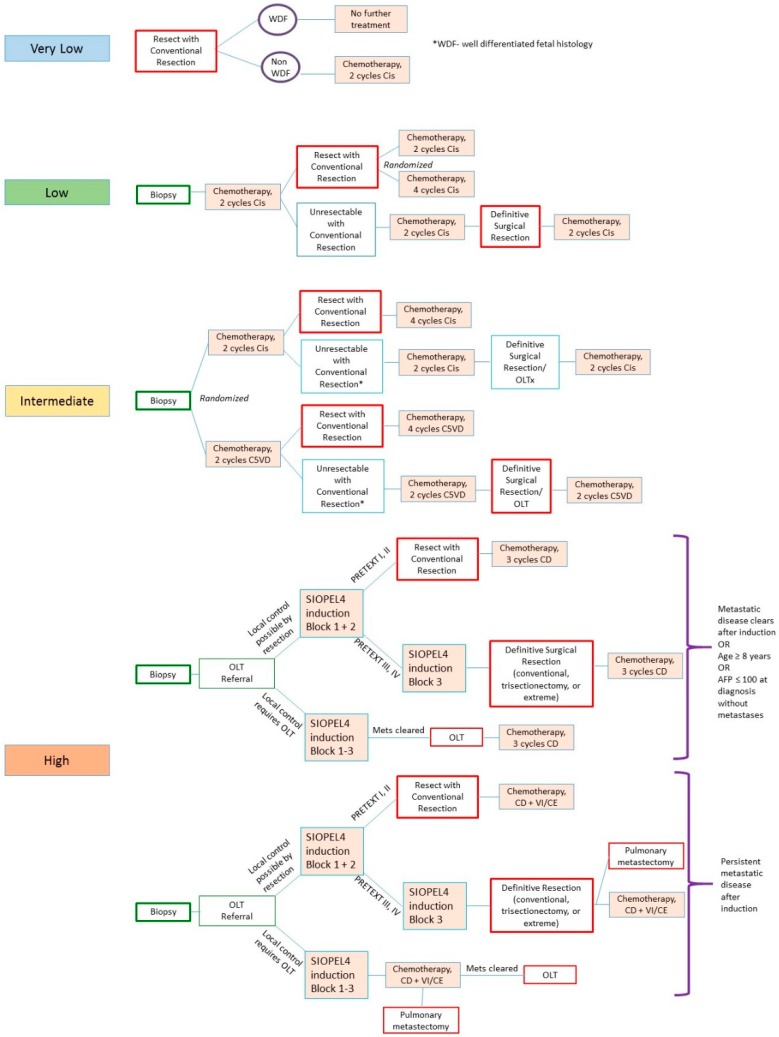
Treatment strategy under AHEP-1531. Management varies based on risk (very low to high) and PRETEXT stage. Cis = cisplatin; C5VD = cisplatin, 5-fluorouracil, vincristine, doxorubicin; CD = carboplatin, doxorubicin; VI/CE = vincristine, irinotecan, or carboplatin, etoposide; WDF = well-differentiated fetal; SIOPEL = Société Internationale d’Oncologie Pédiatrique.

**Figure 4 children-06-00001-f004:**
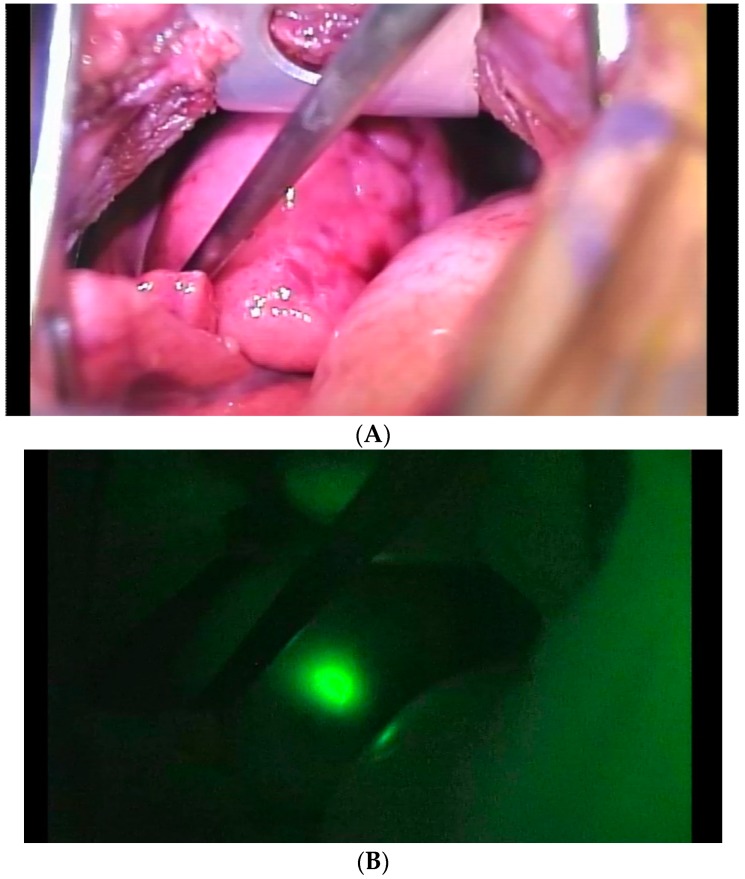
Indocyanine green (ICG) aids in localization of pulmonary metastases. The lesion is indistinguishable from normal lung parenchyma (**A**), but ICG fluorescence clearly delineates the metastatic lesion (**B**).

**Table 1 children-06-00001-t001:** The pre-treatment and post-treatment extent of disease (PRETEXT)/POST-TEXT classification. This classification is used for defining tumors before and after treatment.

PRETEXT/POST-TEXT	Definition
*I*	1 section involved, 3 contiguous sections tumor-free
*II*	1 or 2 sections involved, 2 contiguous sections tumor-free
*III*	2 or 3 sections involved, 1 contiguous section tumor-free
*IV*	4 sections involved, no contiguous sections tumor-free
Subclassifications	
M	Distant, noncontiguous metastasis (i.e. lung)
P	Tumor ingrowth into right and left portal veins, or bifurcation
V	Tumor ingrowth into vena cava, or all 3 hepatic veins
F	Multifocal tumor
C	Tumor in caudate lobe
E	Tumor growth into contiguous organ
R	Pre-diagnosis tumor rupture
N	Lymph node involvement

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
