# Peer review of "Hepatoblastoma—The Evolution of Biology, Surgery, and Transplantation"

_children, 2018, doi:10.3390/children6010001_

Round 1
Reviewer 1 Report
The authors described to introduce treatment algorithms for hepatoblastoma with current improved surgical treatment including liver transplantation, advanced diagnostic imaging and developed chemotherapy regimen. Papers and data were reviewed carefully and concluded properly. And the algorithms they made is simple and useful for physicians who treat children with hepatoblastoma.
Author Response
To Whom It May Concern:
Thank you for your kind comments and review. We have edited the manuscript as follows:
- Clarified the definition of liver sections, as well as defining V and P status in the context of the current trial
- Added Figure 1 as a schematic for the liver sections
- Removed Figure 3
- Corrected for grammatical errors
We again thank you for your consideration of our manuscript.
Regards,
Irene Isabel P. Lim, MD
Alexander J. Bondoc, MD
James I. Geller, MD
Gregory M. Tiao, MD
Reviewer 2 Report
This manuscript well reviewed biological and surgical aspects of hepatoblastoma. However, authors should revise some points to make the manuscript acceptance.
According to surgical aspects, Figure 3 and 4 are nothing new and a quality of Figure 3B is not good. I recommend you to delete or replace those figures with others.
Author Response

(The authors gave the same response as above.)

Reviewer 3 Report
This is an excellent review of an important topic. It is thorough and well-referenced. I have a few suggestions: 1. IN the section that describes the PRETEXT classification scheme, I think liver "section" should be more clearly defined, perhaps with an illustration. We all know what a liver "lobe" is and what a liver "segment" is but the concept of "section" is, I believe, something the folks who developed PRETEXT defined but it might not be intuitive to the average reader. 2. Likewise int he PRETEXT section, the definition of "P" and "V" are not clear as some of us have been taught that it is a P or V if the surgeon does not feel they can achieve at least a 1cm margin from the PV bifurcation or IVC but the text describes it as "involvement" which suggests invasion or encasement. 3. on page two where it describes the various biopsy techniques that are available it suggests that this is up to the "surgeon's discretion" but I would disagree. Naturally, one should always go with the technique that is most likely to provide a. an adequate amount of tissue in the b. safest and c. least invasive way. This is usually by percutaneous biopsy. Open biopsy practically guarantees a large amount of viable tissue but is associated with a significant risk of bleeding. and 4. I find Figure 3 confusing and perhaps not very useful to the average reader.
Thanks again for submitting a very fine paper that I believe is a nice review of an important topic.
Author Response

(The authors gave the same response as above.)
